# Analysis of Hospital Safety and Risk of Falls in the Elderly: A Cross-Sectional Study in Brazil

**DOI:** 10.3390/ijerph21081036

**Published:** 2024-08-06

**Authors:** Leane Macêdo de Carvalho, Letície Batista Lira, Lairton Batista de Oliveira, Annarelly Morais Mendes, Francisco Gilberto Fernandes Pereira, Francisca Tereza de Galiza, Lívia Carvalho Pereira, Ana Larissa Gomes Machado

**Affiliations:** 1School of Nursing, Federal University of Piauí-UFPI, Teresina 64049550, Brazil; leanemacedo79@gmail.com (L.M.d.C.); leticiebl@yahoo.com.br (L.B.L.); 2School of Nursing, Graduate Program in Nursing (PPGENF), Federal University of Piauí- UFPI, Teresina 64049550, Brazil; lairtonoliv@outlook.com (L.B.d.O.); annarellymorais1@gmail.com (A.M.M.); profgilberto@ufpi.edu.br (F.G.F.P.); franciscatereza@ufpi.edu.br (F.T.d.G.); liviacpereira4@gmail.com (L.C.P.)

**Keywords:** elderly, fall accidents, patient safety

## Abstract

This study analyzed hospital safety and the risk of falls in elderly people in a university hospital in Brazil. The Morse Falls Scale was used to stratify the risk of falls in 45 hospitalized elderly individuals, and two checklists were used to analyze the hospital environment. The analysis was based on the Chi-square test and multiple regression. The moderate risk of falls was predominant (51.1%). The variable age group (*p*-value = 0.024) showed statistical evidence of association with the risk of falls. However, the multiple regression analysis showed no difference between the age groups and the risk situation for falls. The hospital wards showed an adequate arrangement of furniture, but some aspects had inadequacies, such as objects in the corridors, non-functional bells in some beds, inadequacy of the toilet bowls in terms of the recommended height, and an absence of non-slip flooring and the support bar in some bathrooms. In conclusion, the moderate risk of falls among the elderly and the adequacy of the hospital environment to technical standards were evident with the exception of failures in the emergency communication system and sanitary installation.

## 1. Introduction

Population aging is a demographic phenomenon that has been increasing for a few decades worldwide. This trend is also noticeable in Brazil, since the share of people aged 60 or over in the country went from 10.8% to 15.8%, showing an increase of 46.6% [1], demanding new arrangements in the organization of healthcare networks due to the increased demand for these services [2,3].

During the aging process, changes occur in the body that are influenced by internal and external factors [4]. The predominant damage to health in elderly individuals is traumatic and related to clinical conditions, directly associated to underlying diseases, such as cardiovascular and metabolic diseases, among others. Moreover, regarding traumatic health problems, falls are the main events [5].

Fall-related health problems in the elderly rank first among the causes of hospitalization, representing approximately 56.1%, and they are the third leading cause of death for external reasons in patients over 60 years of age [6]. In the hospital environment, the rates vary from 1.4 to 13 falls for every thousand patients per day, and damage to patients occurs in approximately 30 to 50% of cases, ranging from minor damage, such as abrasions and bruises, to serious damage, such as femur and hip fractures and head trauma. In the most extreme cases, these falls can even lead to the patient’s death [7].

According to the World Health Organization (WHO), a fall can be understood as an event characterized by the sudden descent of the body to a level lower than the initial position [8]. Additionally, patient falls contribute to prolonging the length of hospital stay, resulting in additional care costs and potentially having consequences for the institution’s credibility and legal issues [9].

Falls can be influenced by intrinsic factors, such as advanced age, previous history of falls, clinical conditions and drug therapy, or by extrinsic factors related to the environment, such as insufficient lighting, slippery carpets, slippery floors and unexpected obstacles [10]. The present research was therefore carried out to investigate intrinsic and extrinsic risk factors for falls, as the assessment of individual risk allows the identification of specific factors that may contribute to the occurrence of falls, allowing the implementation of personalized preventive measures. Moreover, a safe hospital environment with adequate infrastructure and stringent safety protocols significantly reduces the incidence of falls. The combination of these approaches not only improves the quality of care but also promotes faster and safer recovery of elderly patients, minimizing complications and promoting confidence and safety during the hospitalization period.

To promote the safety of patients treated in health units, the Brazilian Ministry of Health, through Ordinance number 2095 of 2013, approved the “Basic Patient Safety Protocols”, which must be applied throughout the Brazilian territory [11]. Among them, the “Fall Prevention Protocol” stands out, which aims to reduce the occurrence of patient falls in care settings and the resulting damage. It is essentially based on the implementation of measures that include patient risk assessment, allowing multidisciplinary care in a safe environment, in addition to promoting patient, family and professional education [12].

There is an agreement between Brazilian regulations and global guidelines for preventing and managing falls in the elderly [13], highlighting the importance of implementing methodologies that analyze the risk of falls through a person-centered approach, taking into account the perceptions of the elderly individual and their interaction with caregivers/companions. Moreover, it is essential to highlight that investment in education aimed at preventing this incident is a low-cost but highly effective strategy.

Thus, given the greater vulnerability of elderly people to falls in hospital environments, it becomes mandatory to carry out this investigation into the intrinsic and extrinsic factors that contribute to this adverse event, especially in hospitals in Brazil where the physical structure is a variable that has aroused the interest of health professionals and managers [14]. Moreover, although the national literature has addressed the topic [15,16], it is known that the physical structure of Brazilian hospitals is widely diverse with precarious architectural resources, which makes it necessary to customize research in the area to strengthen the production of evidence, its application and the development of health policies.

In this context, the present study aimed to analyze hospital safety and the risk of falls in elderly people in a university hospital in Brazil.

## 2. Materials and Methods

The study had a cross-sectional design [17], with data collection being carried out in November and December 2023, in a teaching hospital in northeastern Brazil.

The study population was a census and consisted of elderly patients hospitalized in the following units: Surgical Unit (CIR), High Complexity Oncology Unit (UNACON), Internal Medicine Unit (CM) and Medical/Surgical Clinic Unit (MEDC), totaling 54 patients during the data collection period. The inclusion criteria for the study consisted of being 60 years of age or older and being hospitalized in the units of interest for the study, excluding elderly patients who had hearing impairment, aphasia and those who were disoriented.

The reasons for sample losses include elderly individuals who were not receptive to dialogue, hearing impaired and aphasic individuals, those who were disoriented and absent from bed at the time of data collection. Thus, the study sample was intentional and non-probabilistic, with a confidence level of 95% and an error of 6.02%, comprising 45 elderly patients considering the sample losses.

It is noteworthy that the participants were interviewed individually using the Morse Falls Scale [18] aiming to analyze the individual intrinsic factors that may favor the occurrence of falls. The Morse Fall Scale is used to stratify the risk of falling, considering six parameters: history of falls, secondary diagnosis, support for moving/walking, intravenous medication/heparinized catheter, gait and mental state. Each criterion or parameter corresponds to a specific numerical value, which when added together results in a score, against which stratification is assigned, being classified as “low”, “moderate” and “high” risk of falls. It is important to note that the parameters were investigated considering the participants’ responses according to the guidelines of the original version of the scale [18] and the version translated into Portuguese and validated in Brazil [19], which recommend considering the patient’s self-assessment for each item assessed by the scale.

To analyze the physical hospital structure of the units selected for the study, the checklist for evaluating the environment for individual use (rooms or wards) and the checklist for the physical structure for collective use (internal circulation areas and sanitary facilities) were used [20,21]. It is worth noting that the checklists were extracted from NBR 9050/2020 of the Brazilian Association of Technical Standards (ABNT) [21].

Regarding the analysis of the individual use environment, related to the room or ward, the following safety variables were assessed: presence of bars on the beds, lock on the wheels, bell, auxiliary light, lock on sliding furniture, easy access to belongings, free access to the bathroom, use of non-slip shoes and furniture organized around the bed [20,21]. The variables of internal circulation and sanitary facilities were considered for the structural analysis of the collective environment [20,21].

The data were collected in the rooms/wards of each hospitalization unit as well as in internal circulation areas and sanitary facilities. They were subsequently typed into Microsoft Excel^®^ spreadsheets, version 16.0, double-checked and exported to the Statistical Package for Social Sciences^®^ (SPSS), version 22.0, for statistical analysis (IBM, Armonk, NY, USA). To establish the significance of individual predictors, a univariate analysis was performed using Pearson’s Chi-square test. For characteristics that showed *p* < 0.05, multiple regression analysis was performed with Wald’s Chi-square test, considering a *p*-value < 0.05, Odds Ratio (OR), and 95% Confidence Interval (95%CI).

The study followed all ethical and legal principles for research with human beings [22]. Approval by the Ethics Committee of the University Hospital—HU UFPI—was obtained under Opinion number 6,462,335, and the participants signed the Free and Informed Consent Form (TCLE).

## 3. Results

The average age of the participants was 70.4 years with a predominance of the age group of 60 to 70 years (62.2%). As for the gender, 51.1% were male. In relation to the level of education, there was a predominance of incomplete elementary education with 55.6% of responses.

Among the main reasons for hospitalization reported by the interviewed elderly individuals, pain (26.7%) and surgical treatment (24.4%) stand out. Regarding the self-reported diseases, high blood pressure was the most common among those investigated, totaling 40% of cases. As for the average length of hospital stay, it was estimated at 16.7 days.

Table 1 shows the responses to the Morse Falls Scale items, which comprise the individual fall risk assessment. Regarding the history of falls, 88.9% of patients reported not having suffered a fall in the previous year. As for the secondary diagnosis, 88.9% were diagnosed with more than one disease.

In the item “Support for moving/walking”, the majority of participants (86.7%) were classified as “None/bedridden/nurse support”. Regarding “Intravenous medication”, 60.0% were receiving intravenous therapy. With regard to gait, it was highlighted that 57.8% of those investigated were classified as “Normal/bedridden/wheelchair”. The majority of the interviewees were aware of their capabilities (91.1%).

When evaluating the general fall risk classification scale, it was observed that 51.1% of the elderly had a moderate risk of falls, which was followed by 28.9% with low risk and 20.0% with high risk.

Table 2 analyzed the association of patient characteristics according to the Morse Falls Scale risk classification to establish the significance of individual predictors in the univariate analysis, which revealed that only the age group variable (*p*-value = 0.024) showed statistical evidence of association. The moderate risk of falls predominated equally between the sexes, with 33.3% in the age group of 60 to 70 years, in elderly individuals with incomplete elementary education (24.4%) and with up to 30 days of hospitalization (44.4%).

Considering that only the age group variable showed statistical evidence of association with the risk of falls, as demonstrated in Table 2, the multiple regression analysis was performed for this variable (age group) in the final model and it was observed that there was no difference between the age groups and the risk situation, as demonstrated in Table 3.

In the analysis of the safety of the hospital environment for individual use, as shown in Table 4, it was observed that in all analyzed environments (rooms/wards), the beds had protective rails and locked wheels, there was an operating individual bedside light, with easy handling, and switches located near the bed.

It is noteworthy that the item related to the bell was identified in the majority of beds (99.3%), but they were not working in any of them. Regarding the assessment of circulation and organization of the environment, sliding furniture, such as a bedside table, did not have locks (100%), and the elderly had easy access to their belongings (100%) and bathrooms (98.6%). In only 2 (1.4%) beds, the passage was not clear due to the presence of garbage containers.

As for the physical structure for collective use in the four evaluated hospitalization units, they were in compliance with the standard: handrails present on both sides of stairs and ramps, as well as friezes and contrasting signaling on the edges of the stairs, guaranteeing unobstructed access. The corridors and passages had a regular, continuous and durable coating, with continuous leveling and no steps. All doors had a minimum opening of 80 cm and the handles were lever-type. The signaling had contrasting colors, and there were signs indicating the risk of falls, such as before areas undergoing maintenance or being cleaned.

Among the items that did not comply with the technical standards used in the study, the sanitary installations stand out, as despite having support bars on the sides next to the sanitary bowl for transfer, the toilet bowls did not have an adequate height, the floor was not non-slip and there was floor unevenness at the entrance to the bathroom. Furthermore, the fall risk signaling and support bar next to the shower were present in only one of the assessed bathrooms, and not all bathrooms had a shower chair.

## 4. Discussion

Studies that address the occurrence of falls in the elderly in the context of the hospital environment indicates these events still require greater mobilization of care and management strategies to reduce their frequency, especially because the individual factors of senility associated with the fragility of structures and hospital physical structures can result in irreparable damage to the physical and psychological health of this population [23,24].

On the one hand, the results showed homogeneity between the sexes for the risk of falls; on the other hand, there is evidence that females are the group most exposed to hospitalization as a result of falls [25]. Considering the characteristics of the teaching hospital where this study was carried out, considered to have regulated admission, it cannot be said that the main reasons for hospitalization were falls; therefore, the similarity of risk between men and women should be better investigated in this context.

The mean age of the participants was similar to that of a systematic review with meta-analysis, in which older adults (70 to 81 years) constituted the group at highest risk of falls [26]. The literature highlights that the risk of falls is associated with age and their occurrence is greater in elderly individuals over 70 years old, due to the advanced decline in physical, sensory and cognitive functions resulting from aging [27].

Systematic reviews suggest that advanced age is a significant risk factor for falls, particularly due to factors such as impaired balance, gait problems, polypharmacy and previous falls [26,28,29]. Thus, the similarity between the results of the studies presented herein and the findings of this research show that to promote the safety of hospitalized elderly people and reduce the individual risk of falls in elderly people and people with reduced mobility, the following interventions are necessary: perform fall risk assessments at hospital admission and periodically during hospitalization using scales such as the Morse Falls Scale, provide assistive devices such as walkers and canes and ensure that patients know how to use them correctly, regularly review medications to avoid polypharmacy and reduce the use of medications that may increase the risk of falls, implement exercise and physical therapy programs to improve patient muscle strength, balance, and mobility, and educate patients and their families about risks of falls and prevention measures [30].

Low levels of schooling predominated among the participants, highlighting socioeconomic conditions that can influence the health status of the elderly and their families in addition to access to health information and preventive measures regarding falls [31].

A study carried out in three hospitals in the United States considered that the involvement of patients in understanding and implementing measures aimed at reducing the risk of falls is as important as the monitoring of measures by professionals and the availability of a good infrastructure. With the use of information tools such as printed posters, electronic posters and displays at the head of the bed with fall prevention information, there was an 80% adherence in the assimilation of the content and a consequent reduction in the number of falls [32].

The world guidelines for falls prevention and management for older adults [13] reinforces that involving the elderly is essential for the success in preventing falls; therefore, the elderly person’s level of schooling is essential for educational actions to be effective.

The main reasons for hospitalization of elderly people in the analyzed units were pain and surgical treatment, which are conditions that require attention from the multidisciplinary team in terms of promoting educational measures during the period of hospital stay of the elderly person and their caregiver. Regarding the surgical treatment, a survey carried out in a northeastern Brazilian state [33] supports the results of this study, showing that the main causes of hospitalization were surgical procedures. This health scenario results in high hospitalization rates for elderly individuals, prolonging the time spent in hospital beds, which can increase the risk of falls.

Among the diseases that were self-reported by the research participants, high blood pressure was the most common one, and thus it was considered a risk factor for falls in the elderly. High blood pressure can lead to changes in blood circulation and compromise adequate blood flow to the brain, resulting in dizziness and imbalance, thus contributing to the risk of falls [34].

The average hospital length of stay for the elderly in this study was 16.7 days. Previous studies reported that the average length of stay varied between 7 and 15 days [35,36]. One of the factors that prolongs the hospital length of stay is the delay in receiving definitive treatment. This prolonged length of stay can harm the patients’ balance, increasing the risk of falls in the hospital environment [37].

In the analysis of the Morse Falls Scale items, it was observed that 88.9% of those investigated reported not having suffered a fall in the previous year, which is considered a positive factor, given that elderly people who have suffered a fall in the last six months are more prone to occasional falls than those who have not suffered any falls [38].

However, conditions that increase the occurrence of falls were identified, such as secondary diagnosis (88.9%) and the use of intravenous medication (60.0%). The use of medications is considered to increase falls; in addition to being an extrinsic risk factor that occasionally causes psychomotor and physiological changes, it is also an intrinsic factor resulting from the individual’s specificities [39].

It is noteworthy that some favorable points were identified regarding the items’ support for moving and mental state with the majority of participants demonstrating awareness of their capabilities. However, a significant number of participants showed weakened gait (35.6%) and required support (6.7%), contributing to the incidence of falls due to limitations associated with deficits in balance and locomotion, requiring assistance care to prevent possible damage [38].

Although the aforementioned studies analyzed larger samples than the one analyzed in the present study, it is important to point out that the results of this research reinforce the premise that analyzing the individual risk of falls to identify causes and implement continuous improvements is important to create a safer environment and significantly reduce the risk of falls among hospitalized elderly patients, promoting their recovery and well-being.

Moderate and high risk of falls corresponded to more than half of the elderly individuals investigated in this research (61.1%). These findings corroborate the Brazilian study carried out with 284 elderly people hospitalized in different sectors of a university hospital, in which 79.9% of the participants had a moderate or high risk of falls [38], and the study carried out in China with 153 elderly people hospitalized from 2018 to 2020, in which 88.89% of the participants showed a moderate or high risk of falls [40].

Only the age group variable showed statistical evidence of association with the risk of falls, and when using the high risk of falls as a reference, no difference was observed between the age groups, according to the multiple regression analysis. A systematic review with meta-analysis that proposed evaluating 22 factors and their associations with the risk of falls observed that among the analyzed factors, advanced age considerably increases the risk of falling [26]. A prospective study carried out in the United Kingdom with 3298 elderly people, in a multivariate analysis of 17 factors, showed that only advanced age was associated with an increased risk of falls for both sexes [41].

Therefore, investigating the factors associated with the risk of falls in the hospitalized elderly population is a measure that positively contributes to the quality of healthcare, as it allows monitoring the event and developing effective strategies that reduce the occurrence of falls in the hospital environment and ensure patient safety [42].

Falls in the elderly are often attributed to the lack of adequate clinical conditions, which is an unsafe environment or a combination of these factors [43]. Therefore, when evaluating the hospital environment, it is crucial to consider the organization and physical structure of common areas, such as corridors, to ensure spaces that meet the needs of older adults [31].

Nurses and doctors from an oncology care unit in Kansas (USA) reinforce the relevance of adequate physical and material structure for patient safety and to reduce the risk of falls, such as the availability of wheelchairs in quantities that are proportional to the number of at-risk patients; offering walking belts to assist with the stability of frail elderly people; and standardized signaling on walls and floors to inform the risk of falls [43].

Regarding the assessment of the physical structure, both individual and collective, of the studied hospital unit, the majority of facilities comply with the standard [21]. This result has several positive implications, such as ensuring that all patients, including those with reduced mobility or disabilities, have equal access to hospital services, improving the patient experience by providing a more welcoming and safe environment, which can increase satisfaction and trust in the health system, reduce the risk of falls and other accidents, especially among elderly patients and those with limited mobility, and promote the social inclusion of people with disabilities or reduced mobility, ensuring they can access health services without barriers.

Compliance with accessibility standards represents a significant step forward for university hospital organizations in Brazil, promoting the safety, inclusion, and quality of health services. This not only directly benefits patients and the community but also strengthens the position and sustainability of health institutions in the national scenario.

Nonetheless, some irregularities were still identified in the analyzed hospital environments, such as non-functioning bedside bells. Regarding this result, it is recommended that preventive measures be adopted to ensure the safety of elderly patients, as emphasized in the literature [44]: perform regular inspections of bedside bells and other critical equipment to ensure they are in full working order, implement a preventative maintenance program to identify and correct problems before they affect the patients, train nursing and maintenance staff to identify problems with equipment and report them immediately, and establish clear protocols for communication between patients and healthcare staff, ensuring that all needs are quickly met.

In addition to implementing these improvements, hospitals should ensure that essential features such as guardrails, bedside lights and accessible switches are in place to significantly reduce the risk of falls and promote an environment conducive to patient recovery and well-being [44,45]. These measures demonstrate a commitment to patient safety, providing ideal conditions for multidisciplinary care and the comfort of hospitalized individuals [46]. A study carried out in 12 public hospitals in Malaysia [47] disclosed significant extrinsic factors that led to falls in elderly people, including the absence of a transfer bar by the toilet and a bell.

The analysis of the present research found irregularities in the studied units, which can increase the risk of falls in the elderly, such as toilet seats without elevation, slippery floors and uneven flooring at the entrance to the bathroom. Only one bathroom had fall risk signs and support bars near the shower, and not all bathrooms had a shower chair.

In a national study [48] carried out in a hospital in southern Brazil, problems related to the physical structure of the institution were also found, such as the lack of support bars and water leaks in the bathrooms. These findings are in line with a study carried out in Australia [49], which identified slippery floors in bathrooms and the presence of water or body substances on these surfaces as the main causes of falls among the elderly. Corroborating these findings, another study [47] identified irregularities also related to bathrooms with the absence of a support bar in the toilet.

Given this situation of concern identified in the literature and in the present research, it is necessary to highlight the urgency of physical improvements in hospital units [44] such as the installation of height-adjustable toilets or toilets with a standard height recommended for the elderly and people with reduced mobility, and the application of non-slip coatings on the floors of bathrooms and risk areas, as such improvements combined with adequate staff training and the use of technologies can create a safer hospital environment. It is also considered relevant to collect and analyze patient feedback to identify areas for improvement [50] and adjust fall prevention strategies in addition to conducting regular audits of facilities to identify and correct safety issues.

Technological support is recommended for the implementation of intelligent measures in the hospital environment for monitoring and rapid responses to accidents [51]. Fall sensors and alarm systems in critical areas can be listed, as well as assistive technologies, such as automatic lights and smart non-slip mats, which can alert staff to risks, as can be seen in the studies presented below.

To reduce environmental risks that expose elderly people to fear of falling or the fall itself, researchers have supported the development of technologies such as wireless network sensors, which are attached to the patient, detect the patient leaving the bed or chair and send direct signals to the nursing team so that monitoring and preventive measures can be carried out at the earliest possible moment [52,53].

In Iran, a tiered randomized controlled trial involving 33,856 patients from a university hospital assessed the effectiveness of multiple interventions in reducing fall rates in these patients, such as staff training, adequate lighting, supervision of high-risk patients during transfers, use of mobility devices, placement of a bell and security in bathrooms, placing “fall warning” signs above patient beds, encouraging the appropriate use of glasses, hearing aids and shoes, keeping the side rails elevated, and reassessing patients after each fall. As a result, a decrease in the fall rate was observed from 4 per 1000 patient-days to 1.34 per 1000 patient-days [54].

In a case-control study carried out between 2013 and 2019, the costs of inpatient falls and the cost savings associated with the implementation of an evidence-based fall prevention program implemented in 33 medical and surgical units in eight hospitals of two US healthcare systems were analyzed. The software was associated with savings of USD 22 million at the study sites over the five-year study period and prevented 50 excess deaths [55].

A non-randomized clinical trial developed in 14 medical units in New York, between 2015 and 2018, proposed evaluating a toolkit led by nurses and involving patients and family members in the process of preventing falls during hospitalization. A total of 37,231 patients were evaluated; of these, 17,948 were before the intervention and 19,283 were after the intervention, and a significant association was verified with the reduction in falls [56].

In Spain, a quasi-experimental study with a non-randomized control group, carried out in 2015, evaluated the effect of an educational intervention led by hospital nurses on the systematic fall risk assessment in reducing the incidence of falls. For this purpose, an intervention group (n = 303) and a control group (n = 278) were formed. In the analysis, the intervention group was less likely to fall [57].

Therefore, improving the hospital structure can prevent fall events by ensuring that aspects of the hospitalized patient’s environment (including flooring, lighting, furniture and accessories, such as handrails) that can affect the patient’s risk of falling are systematically identified and addressed [58]. Structural factors must be added to the multifactorial assessment and interventions, such as professional education, education and counseling for patients and family members, and strategies based on specific measures [23,30,59,60].

It is therefore observed that scientific evidence and the context of the study itself point to possibilities for intervention to prevent falls in hospitalized patients, especially the elderly, considering intrinsic factors related to the process of aging and becoming ill as well as extrinsic factors highlighted by the structural and functional inconsistencies of the hospital environment. For that purpose, it is important to develop an agenda of training and management activities centered on goal six of patient safety with the specific demands of each health service, promoting effective solutions implemented by all involved agents.

## 5. Conclusions

The results of this study highlight the vulnerability of older adults (70 to 81 years) to the risk of falls with this group being the most affected. The statistical analysis revealed that the age group was the only variable significantly associated with the risk of falls, highlighting the need for specific interventions for the elderly population.

The predominance of a low level of schooling among the participants reflects unfavorable socioeconomic conditions that not only affect the health of older adults but also limit access to health information and preventive measures. This scenario suggests that educational and socioeconomic efforts are essential to improve the quality of life and safety of older adults.

Moreover, conditions that increase the likelihood of falls were identified, such as secondary diagnoses and the use of intravenous medication. A weakened gait and the need for support were common among the participants, highlighting the importance of appropriate care to prevent falls.

It is a matter of concern that more than half of the assessed elderly individuals were at moderate and high risk of falls, reinforcing the urgent need for prevention strategies focused on improving mobility and balance as well as a multifaceted approach that includes medical, educational and socioeconomic support.

The assessment of the physical structure of the assessed hospital unit revealed that most of the facilities are in compliance with Brazilian technical standards, which has several positive implications. This compliance ensures that all patients, including those with reduced mobility or disabilities, have equal access to hospital services. Moreover, a more welcoming and safe environment improves the patient’s experience, increases satisfaction and trust in the health system, and reduces the risk of falls and other accidents, especially among elderly patients and those with limited mobility.

Promoting the social inclusion of people with disabilities or reduced mobility, ensuring that they can access health services without barriers, represents a significant advance for university hospital organizations in Brazil. This promotes the safety, inclusion and quality of the health services offered.

However, some irregularities were identified in the analyzed hospital environments, such as non-functioning bells on the bedside. To ensure the safety of elderly patients, it is recommended to adopt preventive measures, such as the regular maintenance and inspection of all equipment.

Additionally, it is urgent to carry out physical improvements in the assessed hospital units, including the installation of height-adjustable toilets or toilets with a standard height recommended for the elderly and people with reduced mobility as well as the application of non-slip floor coverings to bathrooms and high-risk areas.

Finally, collecting and analyzing patient feedback is important to identify areas for continuous improvement, ensuring that hospital facilities can constantly adapt and improve to meet the needs of all patients safely and efficiently.

## Figures and Tables

**Table 1 ijerph-21-01036-t001:** Distribution of elderly individuals according to the Morse Falls Scale evaluation criteria. Teresina, Piauí. 2024.

Morse Falls Scale Items	N	%
History of falls		
Yes	5	11.1
No	40	88.9
Secondary diagnosis		
Yes	40	88.9
No	5	11.1
Support for moving/walking		
None/bedridden/nurse support	39	86.7
Crutches/Canadian crutches/walker	6	13.3
Intravenous medication		
Yes	27	60
No	18	40
Gait		
Normal/bedridden/wheelchair	26	57.8
Slow gait speed	16	35.6
Needs support	3	6.7
Mental state		
Aware of their capabilities	41	91.1
Overestimates/forgets their limitations	4	8.9

Source: The authors.

**Table 2 ijerph-21-01036-t002:** Association between the risk of falls and the participants’ characterization. Teresina, Piauí. 2024.

Characteristics	Risk			Total	*p*-Value *
	Low N (%)	Moderate N (%)	High N (%)	N (%)	
Sex					0.901
Male	6 (13.3)	12 (26.7)	5 (11.1)	23 (51.1)	
Female	7 (15.6)	11 (24.4)	4 (8.9)	22 (48.9)	
Age range (years)					**0.024**
60–70	7 (15.6)	15 (33.3)	6 (13.3)	28 (62.2)	
71–81	2 (4.4)	8 (17.8)	3 (6.7)	13 (28.9)	
≥82	4 (8.9)	0 (0.0)	0 (0.0)	4 (8.9)	
Level of schooling					0.819
Illiterate	4 (8.9)	10 (22.2)	2 (4.4)	16 (35.6)	
Incomplete Elementary School	8 (17.8)	11 (24.4)	6 (13.3)	25 (55.6)	
Complete Elementary School	1 (2.2)	2 (4.4)	1 (2.2)	4 (8.9)	
Length of hospital stay					0.258
Up to one month	11 (24.4)	20 (44.4)	8 (17.8)	39 (86.7)	
1–2 months	2 (4.4)	3 (6.7)	0 (0.0)	5 (11.1)	
>2 months	0 (0.0)	0 (0.0)	1 (2.2)	1 (2.2)	

Source: The authors. * Chi-square test, with Yates correction, at 5% level. Bold includes the emphasis on the *p*-value to highlight statistical significance.

**Table 3 ijerph-21-01036-t003:** Regression association between the risk of falls and the participants’ age group. Teresina, Piauí. 2024.

Classification of Morse Falls Scale a		B	df	*p*-Value *	OR	OR-95%	
	Age Range (Years)					Lower Limit	Upper Limit
Low Risk	Intercept	18.948	1	0.996			
	60–70	−18.794	1	0.996	6.888 × 10^5^	0.000	b
	71–81	−19.353	1	0.996	3.936 × 10^−9^	0.000	b
	≥82	0 c	0				
Moderate Risk	Intercept	0.938	1	0.166			
	60–70	−0.022	1	0.979	0.978	0.192	4.993
	71–81	0.043	1		1.043	1.043	1.043
	≥82	0 c	0				

Source: The authors. a. The reference category is: high risk. b. Multiple regression analysis, with Wald’s Chi-square test. c. This parameter is set to zero because it is redundant. * *p*-value < 0.05, Odds Ratio (OR), and 95% Confidence Interval (95% CI).

**Table 4 ijerph-21-01036-t004:** Assessment of individual use environment security variables. Teresina, Piauí. 2024.

Variables	Answer	Total Beds Assessed	
		N	%
Beds with protective rails	No	0	0
	Yes	145	100
locked wheels	No	0	0
	Yes	145	100
Bedside light	No	0	0
	Yes	145	100
Nearby bell	No	1	0.7
	Yes	144	99.3
Free access to the bathroom	No	2	1.4
	Yes	143	98.6
Sliding furniture without locks	No	0	0
	Sim	145	100
Organized furniture	No	0	0
	Yes	145	100
Easy access to one’s belongings	No	0	0
	Yes	145	100
Non-slip footwear	No	145	100
	Yes	0	0

Source: the authors.

## Data Availability

The data presented in this study are available on request from the corresponding author.

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
