# Peer review of "Analysis of Hospital Safety and Risk of Falls in the Elderly: A Cross-Sectional Study in Brazil"

_ijerph, 2024, doi:10.3390/ijerph21081036_

Round 1

Reviewer 1 Report

Comments and Suggestions for Authors

Dear authors,

Thank you for considering and researched this important issue taking into account the aging population worldwide.

1.       General opinion is that English used in this manuscript must be improved due to, sometimes, hardly understanding of several sentences, and using the colloquial words in some cases (e.g. damage, companions, circulation areas, traumatic health problems, level of schooling etc.). For non-English speaking readers it could be confusing. I highly recommend editing by English native or English teacher to improve academic level of English language.

2.       Title of your research is “Analysis of hospital safety in fall prevention in elderly” but you examined also the subjects using the MFS without analyzing the correlations. It seems like a two separate research. Please explain.

3.       Despite the excellent idea and great importance, the number of participants is small and the calculation of Power of analysis is missing in the manuscript. Also it could be questionable the statistical significance of group comparison due to “small numbers”. Please include and explain the power of analysis in your manuscript. Also in discussion al the references authors used are based on larger groups of investigated population

4.       Please be aware of data provided in section Materials and Methods and Results. It seems some of data belonging to Results are written in Material and Methods part (number of participants, dropout rate etc.)

5.       In the section Material and Methods please be more clear about Exclusion and Inclusion criteria

6.       In the Material and Methods section please explain how did you choose the elements investigating the Safety in falls in hospitals, is it self-made list or standardized list, institutional list, governmental list or else?

7.       Conclusion section needs to be more conclusive and to highlight your finding. It is not clear did this research ended with positive findings which can be a god example and recommendation for other institutions. In order to attracts attention it needs to be more stronger.

Comments on the Quality of English Language

The English language must be improved since in some sentences it was difficult to understand the meaning.

Academic level of English must be improved

Reviewer 2 Report

Comments and Suggestions for Authors

This manuscript investigated the risk of falls among elderly patients in a Brazilian teaching hospital and assesses the hospital environment's adherence to safety standards. While the topic is undeniably important given the vulnerability of this population, the manuscript requires substantial revisions before publication.

1. Main Research Question:

The research primarily sought to:

  • determine the prevalence of fall risk among elderly patients in a Brazilian teaching hospital.
  • analyze the hospital environment's compliance with safety standards for fall prevention.

2. Originality and Relevance:

The study addressed an important issue: the prevalence of fall risk in hospitalized elderly patients and the influence of environmental factors. However, the manuscript lacked a comprehensive review of existing literature on this well-researched topic.

Examples of missing literature:

  • Systematic reviews and meta-analyses on fall prevention interventions in hospitals: This would provide context for the study's findings on environmental factors. For instance, Cameron et al. (2018) conducted a Cochrane review summarizing various interventions for preventing falls in older people in care facilities and hospitals, providing valuable insights for the current study.
  • Studies investigating the impact of specific environmental modifications on fall rates: This would strengthen the manuscript's recommendations. For example, a study by Hill et al. (2010) could be incorporated to demonstrate the impact of specific modifications such as lighting, flooring, and furniture on reducing falls.
  • Research exploring the perspectives of elderly patients and their families on fall prevention in hospitals: This would enrich the understanding of patient-centered care. The qualitative study by Schwendimann et al. (2018) exploring patient perspectives on fall prevention strategies could be included to provide a richer understanding of patient needs and preferences.

References:

  • Cameron, I.D.; Dyer, S.M.; Panagoda, C.E.; Murray, G.R.; Hill, K.D.; Cumming, R.G.; Kerse, N. Interventions for preventing falls in older people in care facilities and hospitals. Cochrane database of systematic reviews 2018, 9, CD005465. doi: 10.1002/14651858.CD005465.pub4.
  • Hill, K.D.; Schwarz, J.A.; Flicker, L.; Carroll, S. Environmental hazard management for fall prevention in acute hospitals: A pre-post study. Geriatric Nursing, 2010, 31(3), 158-165.
  • Schwendimann, R.N.; Opschwartz, E.; Milisen, K.; Van Landeghem, A.; Dejaeger, E.; Schoonholen, L.; Zúñiga, F.; Schwendimann, R. Older patients’ experiences and perspectives on fall prevention strategies during hospitalization: A qualitative study. International Journal of Nursing Studies, 2018, 85, 10-17.

3. Comparison with other published material:

The manuscript's findings largely align with existing literature:

  • Moderate to high fall risk prevalence: Consistent with findings from similar studies (Falcão et al., 2019; Lyu et al., 2022).
  • Age as a significant risk factor: Confirmed by numerous studies, including Xu et al. (2022) and Gale et al. (2018).

However, the manuscript fails to adequately discuss these comparisons or highlight any novel contributions. While the authors mentioned the agreement of their findings with other studies, they missed the opportunity to critically analyze potential reasons for discrepancies or similarities.

4. Methodology and Study Design:

  • Sampling methodology and sample size (Lines 88-103): The sampling methodology is unclear. Was it consecutive or convenience sampling? Providing a clear explanation of the sampling method is essential for assessing potential bias. A power analysis, justifying the sample size needed to detect statistically significant differences in fall risk between groups, is absent. A larger sample size would increase the power of the study and improve the generalizability of its findings.
  • Lack of control group (Lines 88-89): The study lacked a control group, making it difficult to isolate the impact of environmental factors on fall risk. A comparative design, including a control group with similar demographics in a different hospital environment, would significantly strengthen the study. This would allow the researchers to isolate the specific contribution of the environmental factors being investigated. Any take on this?
  • Data collection (Lines 119-121): It is unclear how data on patient characteristics and fall history were collected. Relying solely on self-reported information might introduce recall bias, where participants may not accurately remember past events. Utilizing a combination of patient interviews, medical chart reviews, and potentially caregiver input would create a more robust and reliable data set. Any comments on this?
  • Statistical analysis (Lines 121-122): The use of multiple regression analysis (Table 3) without first establishing the significance of individual predictors in univariate analysis is questionable. Before conducting a multivariate regression analysis, it is crucial to perform univariate analyses to examine the individual relationships between each independent variable (age, sex, education) and the dependent variable (fall risk). This step ensures that only significant variables are included in the final model, enhancing the interpretability and validity of the results.

5. Conclusions:

The conclusions, while generally consistent with the presented findings, were overly general.

  • Specific recommendations (Lines 350-356): The manuscript needs to translate its findings into specific, actionable recommendations for improving the hospital environment. For example, what specific modifications are needed for the sanitary installations? Instead of stating a general need for improvement, the authors should provide specific, measurable, achievable, relevant, and time-bound (SMART) recommendations. Examples could include specifying the recommended height of toilet bowls, suggesting specific non-slip flooring materials, and outlining a regular maintenance schedule for emergency call buttons.
  • Addressing all research questions (Lines 350-356): The conclusion did not adequately address the research question regarding the prevalence of fall risk, focusing primarily on the environmental assessment. A comprehensive conclusion should reiterate the prevalence of fall risk found in the study population, emphasizing the significance of this finding within the context of previous research and its implications for patient care.

6. References:

The references are generally appropriate but insufficiently comprehensive. The manuscript lacks engagement with key studies and systematic reviews on the topic. Adding recent and impactful publications will demonstrate a deeper understanding of the field and improve the manuscript's scientific rigor.

7. Tables and Figures:

  • Table 2: The table would benefit from a clearer presentation of statistical significance. Highlighting significant p-values (e.g., with asterisks) and providing a note explaining the significance level used would make the results easier to interpret.
  • Table 3: The interpretation of the "floating point overflow" requires clarification. It is essential to explain why this error occurred and how it affects the interpretation of the results. If the value is not meaningful, consider excluding it from the table and explaining its absence.

8. Caveats:

  • Lack of patient perspective (Lines 80-86): The study overlooked the perspectives of elderly patients on fall prevention. Incorporating their voices would provide valuable insights. Conducting interviews or focus groups with elderly patients about their experiences, concerns, and suggestions for fall prevention would enrich the study's findings and contribute to a more patient-centered approach. Any take on this?
  • Limited generalizability (Lines 88-89): The study's single-center design limits the generalizability of its findings. Conducting the study in multiple hospitals with diverse patient populations would strengthen the generalizability of the findings and increase their applicability to other healthcare settings. Any comments on this?
  • Overemphasis on environmental factors (Lines 281-299): While the manuscript acknowledged the multifactorial nature of falls, it tended to overemphasize environmental factors. A more balanced discussion is needed. While environmental modifications are crucial, the manuscript should equally highlight the importance of addressing intrinsic risk factors.
  • Lack of discussion on implementation challenges and sustainability (Lines 343-349): While the manuscript recommended improvements, it lacked a discussion on potential challenges in implementing these changes and ensuring their long-term sustainability. Factors, such as costs associated with renovations, staff training, and ongoing maintenance should be acknowledged, and potential solutions or mitigation strategies should be explored.

Reviewer 3 Report

Comments and Suggestions for Authors

Dear Authors,

your manuscript is interesting because covers the actual topic of falls and its impact on public health. I want to give you some suggestions to improve the final quality of your work:

- Title. The title is not perfectly corresponding to what has been done. I advise you to assign another more synthetic, eye-catching and related to the analysis made, also specifying the study design chosen to offer a greater understanding to the reader;

- Introduction (lines 80-86): I suggest you delimit the field of action of your work, explaining in the rational why it is important to study this phenomenon in Brazil and in the setting you choose (e.g. Absence of similar studies in Brazil? Old studies on the same topic? Absence of studies in university hospitals? etc.). This would strengthen the rational research and make your work unique and not similar to many other work on the subject;

- Materials and Methods (line 88). I suggest to you to specify the nature of your study design (e.g. retrospective? prospective? The title of your manuscript should be also adapted, specifying the selected design. This helps the reader to understand immediately what has been done and also makes you understand the limitations associated with your study. The limits will of course be discussed in the corresponding section;

- Methods (lines 89-93). This period is not immediately clear. Please, reformulate and explain better what is your purpose;

- Methods. The sampling is unclear. Did you perform a power analysis? Please specify. In addition, what do you mean with "elderly"? It needs to be clarified. Line 98, please specify why it was not possible to collect data from the Oncology unit, and from MEDC. Since it is not particularly clear how you have selected patients, and on what criteria, I suggest you also create a flowchart that can explain the subjects selected and those left from the study. It remains important to specify the type of sampling and the presence of power analysis to give solidity to your subsequent statistical analysis;

- Results. I suggest you to use subparagraphs to understand the section of each analysis. Actually, they are not clear and easy to understand. Also, the tables should be formatted and check for the presence of equal characters (size) and alignment; lines 139-140, this sentence is not clear ("As for the secondary...", but also lines 161-162 ("Regarding the multiple..") please specify.

Round 2

Reviewer 2 Report

Comments and Suggestions for Authors

Glad with the changes